# Sentiment Analysis: An ERNIE-BiLSTM Approach to Bullet Screen Comments

**DOI:** 10.3390/s22145223

**Published:** 2022-07-13

**Authors:** Yen-Hao Hsieh, Xin-Ping Zeng

**Affiliations:** 1Department of Business Administration, National Formosa University, Yunlin 632301, Taiwan; 2Department of Information Management, Tamkang University, New Taipei City 251301, Taiwan; zxp1002010@gmail.com

**Keywords:** sentiment analysis, ERNIE, BiLSTM, bullet screen comments

## Abstract

Sentiment analysis is one of the fields of affective computing, which detects and evaluates people’s psychological states and sentiments through text analysis. It is an important application of text mining technology and is widely used to analyze comments. Bullet screen videos have become a popular way for people to interact and communicate while watching online videos. Existing studies have focused on the form, content, and function of bullet screen comments, but few have examined bullet screen comments using natural language processing. Bullet screen comments are short text messages of different lengths and ambiguous emotional information, which makes it extremely challenging in natural language processing. Hence, it is important to understand how we can use the characteristics of bullet screen comments and sentiment analysis to understand the sentiments expressed and trends in bullet screen comments. This study poses the following research question: how can one analyze the sentiments ex-pressed in bullet screen comments accurately and effectively? This study mainly proposes an ERNIE-BiLSTM approach for sentiment analysis on bullet screen comments, which provides effective and innovative thinking for the sentiment analysis of bullet screen comments. The experimental results show that the ERNIE-BiLSTM approach has a higher accuracy rate, precision rate, recall rate, and F1-score than other methods.

## 1. Introduction

In the Internet world, people have the right to publish information. Netizens not only post product reviews, but also express their opinions on everything. Social media has become a medium for people to share about their lives, obtain information, and communicate. Social media enables users to publish content on platforms to share information with other users and to communicate with each other [1]. The content is usually presented as text and reflects the emotions and subjective opinions of users. Commodity suppliers can analyze the sentiment of reviews to identify the satisfaction of user groups and problems related to their products [2]. For example, the service industry can quickly adapt to changing customer needs by examining the consumption experiences and feedback of users. Comments and messages on social media can allow governments and enterprises to understand trends in public opinion and then take appropriate action in response to public opinion [3]. It can be said that comments express users’ most authentic experiences and opinions and thus affect others’ opinions and judgments [4].

At the same time, with the continuous development of online video platforms and the popularity of smartphones, users find it more convenient to watch videos, and the amount of video data and the number of video users are increasing year by year. Due to the increasing competition among major video websites and platforms, traditional video platforms have gradually transformed into a new type of social media. For example, bullet screen video is very popular among young people. Bullet screen comments move from right to left on the video screen and synchronize with the timeline of the video. When watching bullet screen videos, users can send real-time bullet screen comments and share and exchange ideas with other users [5]. As a novel and diverse video commenting platform, bullet screen video not only allows users to express their emotions and thoughts, but also enhances the interaction between users and the enjoyment of commenting and makes it easy for users to respond to the video content or users’ comments [6]. Yan et al. [7] proposed that bullet screen videos provide both user and platform value. When entertainment is the main purpose of watching videos, the content of bullet screen comments meets the leisure and self-expression needs of users. In addition, bullet screen not only increases the popularity of a video commenting platform, but also builds a new virtual ecosystem for the video commenting platform through users’ enjoyment of bullet screen comments and increases users’ platform stickiness.

However, because bullet screen video is a real-time comment system that has emerged in East Asian countries, such as China, South Korea, and Japan, in recent years, other countries have conducted less research on it [8]. To our knowledge, existing research mainly focuses on cultural values and communication methods. Studies have focused on the form, content, and function of bullet screen comments, but few have examined bullet screen comments using natural language processing [9]. Bullet screen comments are short text messages of different lengths and ambiguous emotional information, which makes it extremely challenging in natural language processing [9]. As these comments represent users’ opinions or preferences, they are extremely valuable to both the platform users and the platform. Therefore, it is important to understand how we can use the characteristics of bullet screen comments and sentiment analysis to understand the sentiments expressed and trends in bullet screen comments.

This study hopes to help governments and enterprises to monitor public sentiment in social media by studying the emotional content of bullet screen comments. Therefore, this study poses the following research question: how can one analyze the sentiments expressed in bullet screen comments accurately and effectively? To solve the research question, this study proposes an ERNIE-BiLSTM approach to analyze the sentiments expressed in bullet screen comments. This study adopts pre-trained models in natural language processing and deep learning methods to analyze short text messages and analyze the sentiments expressed in bullet screen comments.

## 2. Literature Review

### Sentiment Analysis

Sentiment analysis is one of the fields of affective computing, which detects and evaluates people’s psychological states and sentiments through text analysis [4]. Sentiment analysis belongs to the category of information retrieval or natural language processing. It is an important application of text mining technology and is widely used to analyze comments [10] and aggregate multimedia content [11]. Sentiment analysis is based on text analysis, and it aims to extract and analyze sentiment features. The analysis process usually includes text extraction, text preprocessing, sentiment information extraction, and sentiment polarity determination [12]. At present, sentiment analysis methods are mainly divided into two categories: dictionary-based methods and methods based on machine learning. The former uses the sentiment dictionary as the external knowledge source and the dictionary’s rules and sentiment functions to determine the sentiment of a text. The latter turns the problem into a classification problem and applies methods such as machine learning and deep neural networks to determine sentiment categories.

Most data are unstructured and can’t be predefined in specific ways, especially texts in articles, social media, documents, or emails. Traditionally, it is difficult for enterprises to analyze, classify and identify all kinds of texts because of time-consuming and high cost. Therefore, sentiment analysis can be regarded as a feasible way to investigate the opinions, emotions, or attitudes of texts for products, services, products, or online reviews [13]. It is useful for enterprises to understand the feedback and the post experiences from consumers via sentiment analysis. Enterprises can clearly recognize the advantages and disadvantages of services and products from the perspective of consumers’ feedback [14].

Besides, sentiment analysis which can be also called opinion mining is to deeply analyze texts in documents, webs, or reviews in order to extract the key meanings and recognize the polarity classifications [13,14]. In other words, the objective of sentiment analysis is to determine what texts in contexts belong to the positive concept or the negative concept [15]. According to the positive and negative analysis results, enterprises easily to continue to enhance the competition and also figure out the shortcomings of services and products. Enterprises can increase the consistency and reduce the error of data based on a systematic and objective analysis standard of sentiment analysis. Meanwhile, enterprises also immediately recognize the negative feedback of customers in real-time via sentiment analysis. The analysis results can help enterprises to efficiently take action to provide customers with suitable services in order to decrease the customer loss rate [16].

Bullet screen videos have become a popular way for people to interact and communicate while watching online videos. Compared to comments in discussion forums or product reviews, bullet screen comments are short texts with a time order and focus. Sentiment analysis focuses on ways to analyze the emotional information contained in bullet screen comments. Liu et al. [17] defined the tag sets of bullet screen comments, added manually classified training data, and used semi-supervised learning to classify bullet screen videos more accurately. He et al. [18] measured the popularity of bullet screen videos (including popular videos, popular bullet screen videos, and the latest videos) to recommend suitable videos and advertisements to viewers. In addition, they used a random forest algorithm to analyze bullet screen comments, predict the popularity of bullet screen videos, and construct an accurate bullet screen video recommendation system [19]. Cui et al. [20] applied an unsupervised valence-arousal word approach to analyze emoticons and symbols in bullet screen comments and identify the sentiment categories of short texts.

To our knowledge, most of the above studies used traditional methods. There is still room for improvement in the efficiency and effectiveness of these methods. At the same time, bullet screen comments contain a large number of unknown words, and traditional methods have limited accuracy for sentiment analysis. Different from previous studies, this study mainly proposes an ERNIE-BiLSTM approach for sentiment analysis on bullet screen comments, which provides effective and innovative thinking for the sentiment analysis of bullet screen comments.

## 3. Methodology

### 3.1. Data Collection and Preprocessing

This study collects appropriate bullet screen comments and conducts three experiments to verify the correctness and effectiveness of the proposed ERNIE-BiLSTM approach for sentiment analysis on bullet screen comments. This study mainly collects bullet screen comments on the Bilibili platform and uses a web crawler method to collect all the relevant bullet screen videos and comments from 6 February 2020 to 6 March 2020. The dataset contains over 20,000 bullet screen comments in all the videos of the one-star action for a Dingding event. Ripley [21] suggested that the training corpus should be divided into a training set, a validation set, and a test set to improve the accuracy of machine learning models and prevent model overfitting, which reduces the accuracy. In this study, comments are divided into about 16,000 training comments, 2000 validation comments, and 2000 test comments in a ratio of 8:1:1. The training set, validation set, and test set include 50% positive comments and 50% negative comments [22]. As using the same dataset for comparative experiments can effectively control variables and reduce the distortion of comparative experiments, this study uses the same dataset for model retraining and testing.

Furthermore, for all the collected bullet screen comments, this study first excludes noisy data (including nonsense text or emoticons) to avoid false sentiment analysis results. This study artificially labels the sentiments of all comments to facilitate model testing. In addition, because Chinese differs from English in terms of syntax, it is necessary to segment Chinese text and form a word sequence with complete and accurate semantic expression. Therefore, this study uses transformer-based pre-trained models for word embedding.

### 3.2. Enhanced Representation through Knowledge Integration

Bidirectional encoder representation from transformers (BERT) is a pre-trained language representation model proposed by Google in 2018. Google uses a large amount of text data and adopts an unsupervised learning method to train the model, which includes numerous transformer encoder modules. The overall structure of BERT is shown in Figure 1, where $E_{n}\;$ represents the input message that passes through *n* modules. If we use the transformer encoder modules as compute nodes, we find that the input of the node of each layer is the output of the previous layer. Figure 1 shows the structure of a two-layer transformer encoder module, and the transformer encoder module between each layer is bidirectional. Last, the output calculated by the two-layer module is $T_{n}$, and Figure 1 shows that each input $E_{n}$ corresponds to one $T_{n}$. The input of BERT is composed of three types of embedding layers, namely token embedding, segment embedding, and position embedding [23]. Token embedding can not only capture the semantic and syntactic information of words but also be applied for the representation of words in the forms of multi-dimensional vectors. Segment embedding is used to distinguish the sentences in the text and determine which sentence the word belongs to. Position embedding uses numbers to label the position of the words from the word embedding [23].

As a pre-trained model, BERT mainly performs two tasks:(1)Masked language modeling (MLM): BERT randomly masks 15% of the words from a sentence, and inputs the vector of the masked words in the hidden layer into the Softmax classification layer. Then, the model replaces 80% of the words with masked symbols, replaces 10% of the words with a random word, and leaves the other 10% of the words unchanged. The model aims to solve the problem of “label leakage” in the traditional bidirectional recurrent neural network model.(2)Sentence prediction: When BERT is applied to sentence prediction, it can calculate the relationship between two sentences, and uses the binary classifier to predict whether the input sentence is true and its label. In the model, 50% of the sentences are true sentences and are labeled as IsNext. The other 50% of sentences are obtained by negative sampling and are labeled as NotNext.

Sun et al. [24] proposed enhanced representation through knowledge integration (ERNIE). Inspired by the masked token of BERT, they added the concept of knowledge integration to the model. ERNIE can mask the semantics of words and sentences to learn the full semantics. ERNIE is composed of transformer encoder modules and knowledge. It uses transformer encoder modules to generate corresponding word vectors and integrates short text and entity-level knowledge into semantics.

The ERNIE model vectorizes the text dataset *T*, unifies the text content $t_{b}$ of sentiment classification into a fixed length $L_{max}$, and converts each text $t_{b}$ in *T* into the form of characters to obtain the sequence of characters $T^{\prime }$ (see Equation (1)); where ${t^{\prime }}_{c}$ represents the *c-*th text, c∈[1, len(T)], *d*$\in \left[1,\;len\left(L_{max}\right)\right]$, $W_{d}$ represent the *d*-th word in each text (see Equation (2)):(1)$$T^{\prime }=\left\{\;{t^{\prime }}_{1},\;{t^{\prime }}_{2},\;\ldots ,\;{t^{\prime }}_{c},\;{t^{\prime }}_{len\left(T^{\prime }\right)}\right\}$$
(2)$${t^{\prime }}_{c}=\{\;W_{1}{,}_{\;}W_{1},\ldots ,\;W_{d},\ldots ,\;W_{L_{max}}\}$$

We input each character of ${t^{\prime }}_{c}$ into ERNIE’s word embedding layer, position embedding layer, and dialogue embedding layer, respectively. Next, we obtain three vectors, namely, V1, V2, and V3, and input the sum of the three vectors into the bidirectional transformer layer of ERNIE to obtain a sequence of word vectors $S_{i}$ (see Equation (3)), where $V\left(W_{e}\right)$ represents the word vector of the *e*-th word:(3)$$S_{i}=\{V\left(W_{1}\right),V\left(W_{2}\right),\ldots ,\;V\left(W_{e}\right),\ldots ,V\left(W_{L_{max}}\right)\}$$

The final output is a sequence S of word vectors consisting of len(T) numbers of $S_{i}$, where $S_{i}$ is the output vector of the *i-*th word:(4)$$S=\{S_{1},S_{2},\ldots ,\;S_{i},\ldots ,S_{len\left(T^{\prime }\right)}\}$$

### 3.3. Bidirectional Long Short-Term Memory

Recurrent neural networks (RNNs) that employ recurrent links among hidden layers were applied in the artificial intelligence field. However, RNN encounters a problem of learning the long-term historical data [25]. A Long Short Term Memory (LSTM) neural network is a particular type of RNN that is more appropriate when it comes to modeling long-range dependencies [26,27]. Besides, LSTM also can effectively avoid exploding and vanishing gradient problems that RNN suffers from during back propagation optimization [28].

LSTM’s architecture contains memory blocks instead of hidden units by comparing to RNNs. A memory block contains one or more memory cells that are modulated by nonlinear sigmoidal gates. These gates determine whether the model keeps the values at the gates (i.e., the gates evaluate to 1) or abandons them (i.e., the gates evaluate to 0). Given the input sequence $\;x=(x_{1}\ldots \ldots x_{T}),$ the network computes a mapping sequence to the output $\;y=(y_{1}\ldots \ldots y_{T}).$ The following equations determine the unit activations:(5)$$i_{t}=\sigma \left(W_{xi}x_{t}+W_{hi}h_{t-1}+W_{Ci}c_{t-1}+b_{i}\right)$$
(6)$$f_{t}=\sigma \left(W_{xf}x_{t}+W_{hf}h_{t-1}+W_{Cf}c_{t-1}+b_{f}\right)$$
(7)$$c_{t}=f_{t}c_{t-1}+i_{t}\tan h\left(W_{xc}x_{t}+W_{hc}h_{t-1}+b_{c}\right)$$
(8)$$o_{t}=\sigma \left(W_{xo}x_{t}+W_{ho}h_{t-1}+W_{Co}c_{t-1}+b_{o}\right)$$
(9)$$i_{t}=o_{t}\tan h\left(c_{t}\right)\;$$
where σ is the logistic sigmoid function; gates *i*, *f*, *o*, and c are the input gate, forget gate, output gate, and cell activation vector, respectively. All these vectors have the same size as the hidden vector *h*. The *W* terms denote the weight matrices from the cell to gate vectors. Here, tan h denotes the output activation function [26,27] (as shown in Figure 2).

Graves et al. [29] proposed using BiLSTM to solve the problem of the traditional LSTM model being unable to process related words in sentences from back to front. BiLSTM fully considers the preceding and succeeding contexts of sentences to extract bidirectional semantic features. Siami-Namini et al. [30] found that the BiLSTM model has greater predictive power than the LSTM model. Kim and Moon [31] proved that the BiLSTM model outperforms the LSTM model in processing multivariate time series data through experiments. For example, the vector
$\left[w_{1},w_{2},w_{3},w_{4},w_{5}\right]$ represents a sentence. The input vector of the forward LSTM is $\left[w_{1},w_{2},w_{3},w_{4},w_{5}\right]$, and the input vector of the backward LSTM is $\left[w_{5},w_{4},w_{3},w_{2},w_{1}\right]$. Next, we extract and integrate the feature vectors from the two to calculate the vector of the bidirectional LSTM. The vector obtained from the output of the BiLSTM model is shown in Equation10, where $\vec{h_{i}}$ represents the output of the forward LSTM, and $\overset{\leftarrow }{h_{i}}$ represents the output of the backward LSTM.
(10)$$h_{i}=\left[\vec{h_{i}}\;⊕\;\overset{\leftarrow }{h_{i}}\right]\;$$

After using the pre-trained model to train the text vector and using the BiLSTM model to extract the feature vector, we obtain the final vector, but this vector cannot represent the sentiment of the text. This study selects the Softmax function as the sentiment classifier (see Equation (11)), where $\hat{h_{i}}$ is the predicted probability of “positive” and “negative” labels after the normalization of the BiLSTM feature vector set $h_{i}$. When the value of the positive label is close to one, the text is expected to express a positive sentiment. When the value of the negative label is close to one, the text is expected to express a negative sentiment.
(11)$$\hat{h_{i}}=\mathrm{softmax}\left(h_{i}\right)$$

## 4. Experiment Validation and Results

### 4.1. Experiment 1: Experiment Parameter Settings

#### 4.1.1. Experiment Objective

This study proposes an ERNIE-BiLSTM approach for a sentiment analysis of bullet screen comments, for which it is necessary to use pre-trained models and a BiLSTM network model. It is important to set an appropriate number of hidden layers and neurons for artificial neural networks. One hidden layer is the best option under certain conditions [32]; as too many hidden layers may increase the difficulty of training, the model is difficult to converge [33] Therefore, this experiment finds the optimal parameter settings, including the number of hidden layers, the number of neurons in the hidden layer, and the dropout value, to obtain better experimental results.

#### 4.1.2. Experimental Design

First, this study uses the preprocessed dataset and pre-trained models to segment the text and calculate the text vector. Next, the study inputs the obtained text vector into the pre-set BiLSTM network model and tests the parameter settings to find the best parameter setting (Table 1) [34,35].

##### Settings of the Number of Hidden Layers and Neurons

Artificial neural networks can handle simple binary classification problems with only one or two hidden layers. As the proposed sentiment classification is a positive and negative binary classification problem, this experiment tests the settings of the number of hidden layers (i.e., one layer and two layers) and comprehensively evaluates the processing efficiency. The number of neurons in the hidden layer can be calculated by Equation (12) [36]:(12)(α + β)/ 2
where α is the number of neurons in the input layer, and β is the number of neurons in the output layer. In this study, the number of neurons in both the input layer and output layer is 384 [34,35]; hence, the number of neurons in the hidden layer is also set to 384.

##### Settings of the Dropout Rate

Dropout refers to the temporary removal of some artificial neural network units based on a certain probability in a deep learning network. It refers to finding a thinner network from the original network so that a neural unit and a randomly selected neural unit can work together to remove joint effects between neurons and increase the model’s processing power [37]. The dropout rate should first determine whether model overfitting occurs, so we first evaluate whether overfitting occurs in our experiment using a dropout rate starting from zero. If overfitting occurs, we increase the dropout rate by 0.2 until overfitting does not occur to obtain the best dropout rate (maximum value is one).

This study applies four evaluation metrics as the measurement standards of this experiment to evaluate the performance of the artificial neural network model, including accuracy rate, precision rate, recall rate, and F-score [38,39,40]. True positive (*TP*) means that the predicted sentiment and the actual sentiment are both positive. False positive (*FP*) means that the predicted sentiment is positive, but the actual sentiment is negative. False negative (*FN*) means that the predicted sentiment is negative, but the actual sentiment is positive. True negative (*TN*) means that the predicted sentiment and the actual sentiment are both negative.

The accuracy rate is shown in Equation (13):(13)$$\mathrm{accuracy}=\frac{\left(TP+TN\right)}{N}$$

The precision rate is shown in Equation (14):(14)$$\mathrm{precision}=\frac{TP}{TP+FP}$$

The accuracy rate is the ratio of the number of correctly predicted samples in the sentiment analysis to the total number of text comments. The precision rate is the ratio of the number of correctly predicted samples in the sentiment analysis to the number of samples predicted to be positive (including the samples that are predicted to be correct but are actually wrong). The difference between the two is that the accuracy rate considers all the samples, while the precision rate only considers the samples that are predicted to be positive. Therefore, the accuracy rate can directly measure the proportion of correct predictions for all the samples in the sentiment analysis. The precision rate can be used to accurately measure the proportion of correct predictions for all positive samples in sentiment analysis.

The recall rate is shown in Equation (15):(15)$$\mathrm{recall}=\frac{TP}{TP+FN}$$

The recall rate refers to the ratio of the number of text sentiments to the actual number of sentiment classifications, and it is used to measure the reliability of the model’s prediction.

The F1-score is shown in Equation (16):(16)$$F1=\frac{2\times precision\times recall}{precision+recall}$$

The F1-score is the harmonic mean value of the precision rate and the recall rate. It is used to measure the two indicators’ performance.

#### 4.1.3. Experimental Results

In this experiment, the output layer is divided into one layer and two layers, and the dropout rate is divided into six values of 0, 0.2, 0.4, 0.6, 0.8, and 1. Each model has 12 combinations and 12 output results, and we need to find the optimal parameter settings, as shown in Table 2 and Table 3. The BERT-BiLSTM approach performs best with a dropout rate of 0.6 and hidden layers being one or two. Meanwhile, the ERNIE-BiLSTM approach performs best with two hidden layers and a dropout rate being 0.4 or 0.6. Defining the same conditions (i.e., parameters) for different approaches is an essential key to ensure realizable and valid comparison results [41]. We therefore set the above parameters (i.e., two hidden layers and a dropout rate of 0.6) as the optimal parameters of the two methods.

### 4.2. Experiment 2: Efficiency Comparison of Sentiment Analysis

#### 4.2.1. Experiment Objective

This study proposes an ERNIE-BiLSTM approach that applies special pre-trained models to train the word vectors and then inputs the trained word vectors into a BiLSTM network model to extract text features for the sentiment analysis. This experiment mainly compares the proposed ERNIE-BiLSTM approach with other common methods (BERT-BiLSTM and Word2Vec-BiLSTM) to verify the short-text sentiment analysis’s performance. As the above method is based on pre-trained models and a BiLSTM network, we use the variable-controlling approach to conduct the experiments.

#### 4.2.2. Experimental Design

We divide the experiment into two parts. First, we test the differences in the performance of the BiLSTM network model in the text sentiment analysis under different word vector models. Second, we test the differences in the performance of the three methods under different feature extraction models. Table 4 shows the parameter settings of this experiment [35].

##### Comparison of Different Word Vector Models

This experiment uses BiLSTM to extract the features of the short text and uses three different models to train the word vectors. Isnain et al. [42] found that when using the word-to-vector model (Word2Vec) as the word vector embedding model and analyzing Twitter short-text messages, they can achieve an accuracy of more than 90% after adopting the BiLSTM method. Therefore, this experiment adopts three-word vector embedding models, including a Word2Vec model, a BERT model, and an ERNIE model and uses accuracy rate, precision rate, recall rate, and F-score as performance evaluation indicators to test and analyze the efficiency of the different methods.

##### Different Feature Extraction Models

This experiment compares the performance of a text sentiment analysis between the BERT approach and ERNIE approach and the proposed BERT-BiLSTM approach and ERNIE-BiLSTM approach.

#### 4.2.3. Experimental Results

##### Performance Comparison of Different Word Vector Models

Word vector models use vectors to represent Chinese text. High-performance word vector models can effectively improve the accuracy of sentiment classification. We use BiLSTM to extract features from the word vectors obtained by three-word vector representation methods and then perform a binary classification using a sigmoid classifier. Table 5 summarizes the performance of the three methods in generating word vectors for the sentiment analysis. From the results, the accuracy rate of the ERNIE-BiLSTM approach with a value of 0.889 is higher than that of the BERT-BiLSTM approach with a value of 0.875 and the Word2Vec-BiLSTM approach with a value of 0.687. The precision rate of the ERNIE-BiLSTM approach with a value of 0.871 is higher than that of the BERT-BiLSTM approach with a value of 0.853 and the Word2Vec-BiLSTM approach with a value of 0.699. The recall rate of the ERNIE-BiLSTM approach with a value of 0.853 is higher than that of the BERT-BiLSTM approach with a value of 0.847 and the Word2Vec-BiLSTM approach with a value of 0.644. Last, the F1-score of the ERNIE-BiLSTM approach with a value of 0.848 is higher than that of the BERT-BiLSTM approach with a value of 0.833 and the Word2Vec-BiLSTM approach with a value of 0.642. This proves that word embedding plays an important role in Chinese natural language processing, and a BiLSTM approach based on pre-trained models has a better processing performance than a BiLSTM approach based on Word2Vec.

##### Performance Comparison of Different Feature Extraction Models

The previous experiment analyzes differences in the performance of different word vector methods under the same BiLSTM model. In this experiment, we compare the BiLSTM network model based on the BERT or ERNIE word vector model with the pre-trained BERT and ERNIE approach to obtain the performance of four complete training models in sentiment classification. The results of the experiment are summarized in Table 6. The accuracy rate, precision rate, recall rate, and F-score of the ERNIE-BiLSTM approach are 0.889, 0.871, 0.853, and 0.848, respectively. Its overall performance is higher than that of the BERT pre-trained approach, the ERNIE pre-trained approach, and the BERT-BiLSTM approach. Therefore, the processing performance of ERNIE with the concept of knowledge representation is better than that of the BERT, and the model that includes ERNIE in the pre-trained models based on the transformer structure performs better in Chinese natural language processing.

### 4.3. Experiment 3: Analysis of Public Opinion Strength

#### 4.3.1. Experiment Objective

After collecting the bullet screen comments on the Bilibili platform, this experiment uses the ERNIE-BiLSTM approach to analyze trends in text sentiment and uses the susceptible-infectious-recovered model (SIR) to further analyze the public opinion strength of the Bilibili platform and determine overall trends in online comments, which can be used for important management decisions.

#### 4.3.2. Experiment Design

Kermack and McKendrick [43] proposed the SIR model to simulate the spread of infectious diseases. Freeman et al. [44] used the SIR model to simulate the dissemination process of public opinion in social networks, and used turning points and the life cycle of a virus to analyze the characteristics of public opinion dissemination. This experiment uses the number of user replies and likes for a video as the basic indicator for the strength of public opinion, and divides users into susceptible, infectious, potential, and recovered groups (Table 7) [45].

#### 4.3.3. Experimental Results

Figure 3 and Figure 4 show that the number of infectious people was higher than the number of susceptible people from 1 February to 16 February. This indicates a large number of negative comments, which is not normal for an enterprise. As the number of infectious people increased, the number of communicators increased gradually, but the number of recovered people remained small. This showed that the trend in public opinion trend was developing in a direction that was not favorable for Dingding. On 3 February, the number of negative comments grew rapidly, and the growth rate of positive comments was far less than that of negative comments. On 5 February, communicators appeared and the number of negative comments started to increase. On 7 February, recovered people gradually appeared. This is mainly because when there are a lot of negative comments, commenters are more likely to want to express a different opinion. However, after 11 February, the growth rate in the number of recovered people was extremely slow, and the growth rate of communicators remained stable. Therefore, the overall trend was not positive for Dingding.

After 16 February, once Dingding had released its official public relations video, the number of susceptible people gradually exceeded the number of infectious people—susceptible people can be regarded as people whose stance was biased toward Dingding. The growth rate of negative comments gradually declined and became negative, and the growth rate of positive comments was higher than that of negative comments. At the same time, the number of communicators continued to show a large negative growth rate, and as the number of positive comments increased, the growth rate of recovered people became higher than that of communicators. This shows that the public opinion was full of comments that refuted the communicators’ opinions, indicating a gradual improvement in the public opinion of the one-star action for the Dingding event. Dingding’s official public relations video thus played a role in influencing the direction of public opinion.

This study finds that communicators play a key role in influencing public opinion. When the growth in the number of communicators remains high, the strength of public opinion forces companies to face the pressure of negative comments. When the growth rate in the number of communicators is higher than 66%, it means that the trend of public opinion on the event is gradually deteriorating, and enterprise managers should pay attention to this to ensure a timely response to manage public opinion.

## 5. Discussion

In this study, the pre-trained models of ERNIE segment texts by word. These models differ from traditional methods that require Chinese word segmentation and do not need to refer to a frequently updated dictionary. At the same time, pre-trained models and retraining can use the text vectors derived from context to solve the problem of unknown words. This study also uses BiLSTM to divide the text vectors obtained from pre-trained models and uses forward LSTM and backward LSTM to extract features. Therefore, it can accurately determine the sentiment of texts with contrasting transition words. From the above-mentioned experiments, we find that the processing performance of the BiLSTM approach with pre-trained models is better than that of the Bi-LSTM approach without pre-trained models and the BiLSTM approach with traditional word representation. Therefore, the proposed sentiment analysis model is effective for the sentiment analysis of short text comments with a large number of contrast transition words and new words. In terms of performance evaluation indicators, such as accuracy rate, precision rate, recall rate, and F1-score, the proposed model performs better than other similar natural language processing models, and has certain advantages in public opinion analysis on social platforms.

The sentiment of user comments is the most important factor for public opinion pre-warning. The negative sentiment of comments affects the strategy of a company, and the positive sentiment may indicate the degree of user satisfaction with the behavior of a company. The sentiment of user comments allows a company to better understand the current public opinion regarding it and accordingly adjust its strategies or behaviors, thereby reducing the probability of large-scale public opinion incidents and their impact on the company. This study finds that sentiment in user comments is affected by many factors, such as the intention of the videos, the trend in opinion leaders’ comments, and the sentiment of popular comments. These factors can increase the accuracy of public opinion analysis. The characteristics of changes in public opinions obtained from past public opinion incidents give enterprises sufficient time and data to respond to public opinion events [46].

## 6. Conclusions

Traditional word vector representation methods and recurrent neural networks have a few shortcomings in natural language processing. Traditional word vector representation methods have difficulties in dealing with situations where the context has different meanings (such as Word2Vec) and recurrent neural networks have vanishing gradient and gradient explosion problems. Although LSTM addresses vanishing gradient problems, it can only process in one direction. This study adopts the BiLSTM method, combines forward LSTM and backward LSTM in the text sentiment analysis, solves the problem of semantic processing in one direction, and uses the pre-trained models of ERNIE to effectively improve the accuracy of sentiment analysis.

This study proposes an ERNIE-BiLSTM approach for sentiment analysis on bullet screen comments. After collecting bullet screen comments from the Bilibili platform, this study preprocesses the data to discard meaningless data. First, this study uses ERNIE to encode word vectors to represent the semantics of short texts. It uses BiLSTM to extract text features, deeply analyze text semantics, and complete the sentiment analysis of bullet screen comments. The experimental results show that the ERNIE-BiLSTM approach has a higher accuracy rate, precision rate, recall rate, and F1-score than other methods.

Last, this study conducts an integrated analysis of the one-star action for a Dingding event through the experiment on public opinion strength and uses the analysis and pre-warnings of public opinion to help enterprises understand development trends in public opinion about an event and responses to an event. In addition, all short texts in the datasets used in our experiments were within 128 characters. If the model needs to process long texts with more than 300 characters, the knowledge concepts and graphs contained in the text will be more complex. The overall performance of the ERNIE model is still significantly better than that of the BERT model with pre-trained data of the same size.

We suggest several directions for further research. As the sentiment category of the dataset is not just positive and negative, it can include sentiments such as joy, anger, and sadness to improve the accuracy of the analysis. In addition, in the subsequent complex text sentiment analysis, if the aspect concept can be considered, the applicability of the sentiment analysis model will be effectively improved.

## Figures and Tables

**Figure 1 sensors-22-05223-f001:**
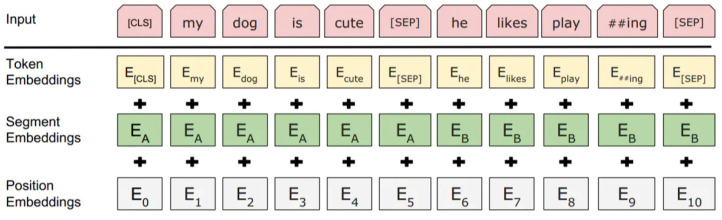
The structure of BERT [23].

**Figure 2 sensors-22-05223-f002:**
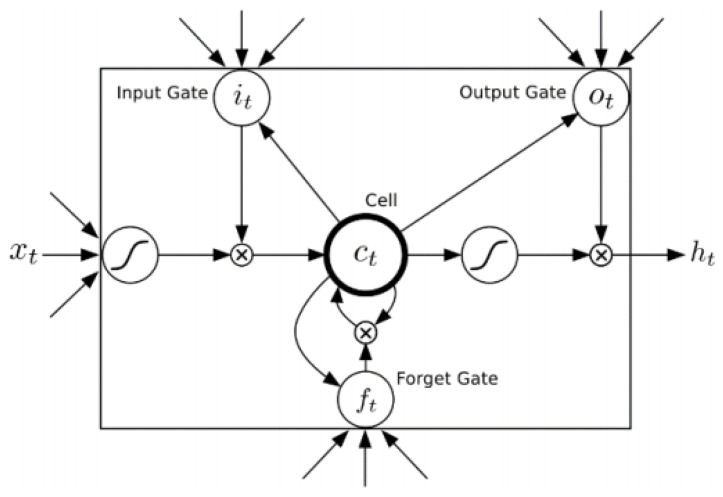
Long Short—Term Memory Neural Network [26].

**Figure 3 sensors-22-05223-f003:**
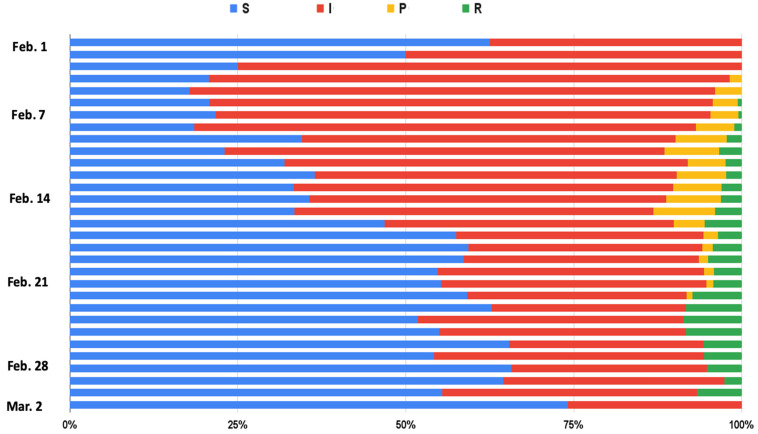
The variations of SIR.

**Figure 4 sensors-22-05223-f004:**
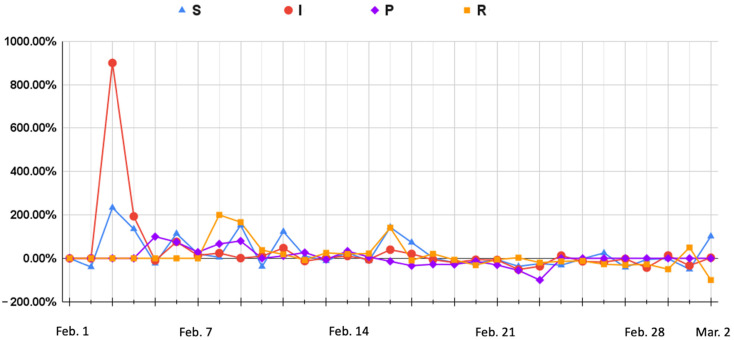
The trends of SIR.

**Table 1 sensors-22-05223-t001:** Parameters Setting.

Parameter	Values
Learning Rate	2 × 10^−5^
Epoch	10
Optimizer	Adam
Numbers of Neurons	384
Batch Size	32

**Table 2 sensors-22-05223-t002:** Experiment results of BERT-BiLSTM approach.

Number of Layer	Dropout Rate	Accuracy	Precision	Recall	F1-Score
1	0	0.832	0.811	0.795	0.789
1	0.2	0.852	0.843	0.824	0.817
1	0.4	0.861	0.851	0.844	0.831
1	0.6	0.872	0.862	0.847	0.841
1	0.8	0.838	0.813	0.801	0.791
1	1	0.513	0.454	0.419	0.407
2	0	0.841	0.819	0.804	0.797
2	0.2	0.855	0.833	0.812	0.809
2	0.4	0.860	0.851	0.842	0.825
2	0.6	0.875	0.853	0.847	0.833
2	0.8	0.836	0.819	0.808	0.801
2	1	0.483	0.471	0.433	0.420

**Table 3 sensors-22-05223-t003:** Experiment results of ERNIE-BiLSTM approach.

Number of Layer	Dropout Rate	Accuracy	Precision	Recall	F1-Score
1	0	0.827	0.801	0.779	0.771
1	0.2	0.847	0.835	0.815	0.796
1	0.4	0.863	0.851	0.844	0.841
1	0.6	0.859	0.849	0.838	0.829
1	0.8	0.827	0.806	0.793	0.781
1	1	0.493	0.478	0.436	0.428
2	0	0.839	0.837	0.821	0.814
2	0.2	0.868	0.867	0.843	0.835
2	0.4	0.873	0.868	0.859	0.853
2	0.6	0.889	0.871	0.853	0.848
2	0.8	0.860	0.859	0.839	0.822
2	1	0.513	0.487	0.445	0.427

**Table 4 sensors-22-05223-t004:** Parameters Setting of Word2Vec.

Parameter	Values
Learning Rate	0.002
Epoch	10
Optimizer	Adagrad
Numbers of Neurons	384
Batch Size	32

**Table 5 sensors-22-05223-t005:** Performance Comparison of Different Word Vector Models.

	Accuracy	Precision	Recall	F1
Word2vec + BiLSTM	0.687	0.699	0.644	0.642
Bert + BiLSTM	0.875	0.853	0.847	0.833
ERNIE + BiLSTM	0.889	0.871	0.853	0.848

**Table 6 sensors-22-05223-t006:** Performance Comparison of Different Feature Extraction Models.

	Accuracy	Precision	Recall	F1
Bert	0.821	0.815	0.801	0.789
ERNIE	0.838	0.824	0.810	0.803
Bert + BiLSTM	0.875	0.853	0.847	0.833
ERNIE + BiLSTM	0.889	0.871	0.853	0.848

**Table 7 sensors-22-05223-t007:** Definitions of SIR.

Role	Definition
Susceptible	A user indicated first post belonging to positive sentiment
Infectious	A user indicated first post belonging to negative sentiment
Potential	A user indicated negative sentiment posts with lots of followers and responses
Recovered	A user indicated positive sentiment posts to respond to Potential

## Data Availability

Not applicable.

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
