# Peer review of "Sentiment Analysis: An ERNIE-BiLSTM Approach to Bullet Screen Comments"

_sensors, 2022, doi:10.3390/s22145223_

Round 1

Reviewer 1 Report

The paper proposes the ERIN-BiLSTM approach to bullet screen comments for sentiment analysis. The paper has potential but it requires major revision before acceptance.

1. The paper lacks open questions and motivations in the abstract.

2. The paper requires more literary works related to the sentiment analysis and LSTM:

https://www.hindawi.com/journals/cin/2022/5681574/ for the sentiment analysis

https://arxiv.org/abs/2108.07467 for the application of LSTM.

You may add more papers related to the field and link them with the current work.

3.Tables 5 & 6 are broken

Reviewer 2 Report

This paper presents an ERIN-BiLSTM approach for sentiment analysis on bullet screen comments. The authors compared the performance of their model with the existing models on the binary classification and feature extraction of the comments, which showed that the proposed one is constantly effective. The improvement, however, is insignificant, especially compared with BERT-BiLSTM. The author also used ERNIE-BiLSTM to analyze the trends in text sentiment according to the susceptible-infectious-recovered (SIR) model. Overall, the paper is well organized, but the proposed approach is basically a combination of existing models. The novelty of the paper is low. Concerns:(1) A clearer description of the model, especially the "ERIN" part and its relationship to "ERNIE" should be provided. Is ERIN-BiLSTM same as ERNIE-BiLSTM? Although the authors provided some explanation in the conclusion (lines 462-468), a clear description of the model should be provided early (e.g., right after it is proposed in Section 3).  (2) lines 286-287, the authors claimed that "The BERT-BiLSTM approach performs best with two hidden layers and a dropout rate of 0.6, while the ERIN-BiLSTM approach performs best with two hidden layers and a dropout rate of 0.8;". However, from the results shown in Tables 2 and 3, it seems BERT-BiLSTM performs the best with one hidden layer and dropout rate of 0.6, and ERIN-BiLSTM performs roughly the same with two hidden layers and dropout rate being 0.6. A better explanation should be given.(3) Table 2 and Table 3, should the heading of the first column be "Number of Output" or "Number of Layer"? Also, should it be the number of hidden layers instead of output layers?(4) If this model will be applied for both real-time and offline sentiment analysis of the bullet screen comments, time efficiency of the model should be evaluated for the real-time case. 

Minor format and writing issues:(1) line 37, insert "the" before "increasing competition"(2) lines 59-60, "As a new way of social interaction, bullet screen videos allow users to publish short and informative text while watching videos.", the sentence can be deleted as the content is redundant(3) line 9 and line 78, "It is a key technology for text mining", sentiment analysis is NOT a technology for text mining. It is an application of text mining technology(4) line 478, delete the duplicated "anger"(5) lines 137-142, the inline mathematical expressions have inconsistent format with the text. Please consider using Latex and its in line math mode to reformat the writing. The same problem happens in lines 170-173, lines 177-184, and lines 193-205.

Round 2

Reviewer 1 Report

Accepted. Please proofread the manuscript before publication.

Author Response

Thank you

Reviewer 2 Report

I appreciate the authors' response to address the concerns in the first-round of review. However, some of the concerns remain. Plus, although the experimental results showed that the proposed approach (i.e., ERNIE + BiLSTM) slightly outperformed BERT + BiLSTM, it mainly attributes to the contribution of ERNIE. Overall, the proposed approach is basically a combination of existing models. The research novelty is low. Concerns: (1) lines 331-332, the authors claimed that "The BERT-BiLSTM approach performs best with two hidden layers and a dropout rate of 0.6, while the ERIN-BiLSTM approach performs best with two hidden layers and a dropout rate of 0.6;". However, from the results shown in Tables 2 and 3, it seems BERT-BiLSTM performs the best with ONE hidden layer and dropout rate of 0.6, and ERIN-BiLSTM performs roughly the same with two hidden layers and dropout rate being either 0.6 or 0.4. The results should be better justified.Writing issues and technical problems still exist: (1) line 23 (abstract keywords), "ERIN" should be "ERNIE" (2) Figure 1 is copied from the original BERT paper published in 2018, should replace it with the authors' own example or at least cite the reference.(3) Figure 2 has a similar issue as Figure 1, the picture is from other sources (e.g., https://hub.packtpub.com/what-is-lstm/), should replace it with the authors' own example or at least cite the reference. (4) line 168, "Token embedding splits the text into characters and turns individual characters into words". This is inaccurate! "token embedding" is to represent each token (roughly equals word) using a multi-dimensional vector (e.g., for BERT-base, the dimension is 768). BERT does use WordPiece tokenizer to split text, but it is for achieving a balance between vocabulary size and out-of-vocabulary words.

Author Response

Thank you
